# Citizens' feedback on health service and the responses of health authorities of Bangladesh: An analysis of the Grievance Redress System

**Md Abdullah Saeed Khan**[ID][1]*, **Md Toufiq Hassan Shawon**[ID][2], **Atonu Rabbani**[ID][3,4]

**1** National Institute of Preventive and Social Medicine, Dhaka, Bangladesh, **2** Management Information System, Directorate General of Health Services, Dhaka, Bangladesh, **3** Department of Economics, University of Dhaka, Dhaka, Bangladesh, **4** BRAC James P Grant School of Public Health, BRAC University, Dhaka, Bangladesh

* abdullahdmc@gmail.com

## Abstract

Developing nations like Bangladesh face unique challenges in healthcare delivery system due to resource constraints, making it crucial to assess the responsiveness of healthcare authorities to citizen feedback. This study utilized data from the Ministry of Health and Family Welfare's (MOHFW) Grievance Redress System (GRS) in Bangladesh, collected from January 1st to August 8th, 2023. A total of 11,604 anonymous messages received from health service takers were retrieved and analyzed. Kaplan-Meier analysis, and log-rank tests were conducted to assess feedback response times. Feedback were mainly complaints (60.33%), followed by compliments (22.03%) and suggestions (14.48%). Complaints were primarily related to the health workforce, infrastructure, and service utilization. Responses included forwarding (67.02%) to the relevant department, closure (30.12%), resolution (2.55%), pending (0.05%), and overdue (0.25%). The median response time was 3.48 hours (IQR: 1.26 – 13.00). Kaplan-Meier analysis revealed that moderate and major complaints were significantly less likely to be resolved than minor complaints (p < 0.001). The status and responsiveness of the grievance redress process of the healthcare delivery system in Bangladesh highlighted in this study can be used to plan the enhancement of the redress system and, thereby, improve health service.

## Author summary

Bangladesh faces significant challenges in delivering equitable healthcare due to limited resources. To understand how effectively the system addresses public concerns, we analyzed 11,604 anonymous messages submitted between January 1 and August 8, 2023—through the Ministry of Health and Family Welfare's innovative Grievance Redress System (GRS). Messages fell into three

**Data availability statement:** Data can be freely accessed on the website of Grievance Redress System: https://app.dghs.gov.bd/complaintbox/ However, permission should be sought from the Director, Management Information System (MIS), Directorate General of Health Services, Ministry of Health and Family Welfare, Dhaka, Bangladesh. Contact no: +8801701248005 Email address: mis@director.dghs.gov.bd.

**Funding:** The author(s) received no specific funding for this work.

**Competing interests:** The authors have declared that no competing interests exist.

categories: complaints (60%), compliments (22%), and suggestions (14%). Complaints most often concerned workforce behavior, facility infrastructure, and barriers to accessing services. We found that health authorities responded promptly, with a median response time of 3.5 hours, typically forwarding issues to relevant departments (67%), closing cases (30%), or resolving a small number (2.6%). However, serious complaints were significantly less likely to be fully resolved by the GRS authorities compared to minor ones. Our findings highlight both the responsiveness and the limitations of Bangladesh's grievance system. Strengthening this feedback mechanism—particularly for addressing more serious complaints—could enhance trust in public health services and contribute to improved health outcomes.

## Introduction

The post-industrial world has seen a rapid rise in population around the world. Improvements in technology, medicine, and health care have increased the life expectancy of people around the world. Simultaneously, the healthcare needs of the people also increased. However, health care delivery is riddled with inequity. Nearly half of the people around the world cannot obtain essential health services [1]. Ensuring quality health services is another challenge faced by countries around the world [2]. Quality health service is an integral part of the Universal Health Care related indicators of Sustainable Development Goals (SDG) [2]. Hence, the health systems developed by countries around the globe are striving to deliver quality health services to the people. But developing nations like Bangladesh face a lot more challenges owing to limited resources allocated for health care delivery [3].

Despite the constraints, Bangladesh has been successful in achieving several health-related indicators of SDG through its public health approaches [4]. However, the country's health system is still plagued by numerous shortcomings. Inadequate health workforce, lack of essential medicines, inadequate equipment, lack of cleanliness, insufficient budget, urban-rural inequity, and lack of monitoring and weak accountability are some of the challenges faced by the health service delivery system of the country [3]. According to a 2020 estimate, Bangladesh had only 9.9 healthcare workers per 10000 people, a much lower figure than the median 48.6 global estimate [5]. Moreover, 32% of the healthcare workers were unqualified, making healthcare coordination and delivery a challenging task. Healthcare facilities can provide 50–70% of the medicine needs of patients due to budget constraints and higher loads of patients [6]. Indoor and outdoor patients get only 78% and 39% of the medicine units prescribed, respectively, underscoring the gap between supply and demand of essential medicines. There is a considerable shortage of diagnostic facilities, with regional and urban-rural variations [7]. With plummeting healthcare budgets from 6.02% to 4.04% in 2019, and 67% of healthcare costs being paid through out-of-pocket expenditure [8], healthcare is a costly

undertaking for the citizens of Bangladesh. The huge population size and high demands of healthcare leave the authorities with no option but to prioritize low resources. Hence, patient and providers' feedback is important in identifying and filling the gaps in health service delivery and improving the quality of care. In order to handle the large number of feedback received daily, the health authorities of Bangladesh developed a range of feedback systems operating in the health care centers. This includes written and verbal feedback managed locally, health call centers, and the Grievance Redress System [9].

GRS is a web portal developed and maintained by the Government of Bangladesh to facilitate quick action on grievances of the citizens [9]. Under the Ministry of Health and Family Welfare (MOHFW), this system stores messages received from people who receive and/or provide services in the hospitals through a dedicated mobile number. Based on the contents of the messages, actions are then taken to address the issues raised by the citizens. An analysis of information stored in the GRS could provide an overview of the nature and type of opinions of people regarding the quality, adequacy, and availability of the health services, with their distribution around the country. In addition, the responsiveness of central health authorities to people's feedback can be assessed by evaluating the type and duration of response [10]. Previously, a few studies evaluated this innovative mechanism after its initiation in 2012. Hence, the objective of our study was to examine the types, nature, and distribution of people's feedback received at GRS, and to explore the responsiveness of the health system to client feedback. Our findings would allow an assessment of incorporating this innovative feedback management mechanism in the public health system and inform policymakers about health seekers' satisfaction and grievances to take necessary steps for improving healthcare.

## Methods

### Ethics statement

Formal permission was taken from MIS, DGHS to retrieve data from GRS website. As the data was retrieved from the anonymous records of GRS, no ethical clearance was sought. Informed consent was not required as the data collection procedure did not involve connecting participants.

### Data source, study population, and duration

This study was conducted based on data available from the GRS of the MOHFW, Bangladesh. GRS is an online platform that collects and stores messages sent usually through short messaging service (SMS) to specified phone numbers from hospitals around Bangladesh [9]. These SMS are sent by both the healthcare providers and seekers. The data stored can be freely accessed through the following link: https://app.dghs.gov.bd/complaintbox/. The Management Information System (MIS), Directorate General of Health Services (DGHS) is responsible for the management and maintenance of GRS. After taking formal permission from MIS, we retrieved data stored between 1st January 2023–8th August 2023 from the site. A total of 11604 messages were retrieved. After discarding five messages with missing data, we analyzed 11,599 messages for this study.

### Data structure of GRS system

**Overview.** The GRS system stores the following data in the online system [10]: serial number of the message received, facility name from which the message was sent, data & time of receiving the message, content of the message, source of the message (DGHS and Health Call Center), type of facility, type of feedback, rank, closure type, and closure time. This information is available from the 'Messages' page of the website. There is also an interactive dashboard where a user can find the number and types of messages received per day within a specified duration. Another page is dedicated to a deeper investigation of the messages through detailed graphical analysis.

**Details of the information stored.** The type of facility specifies if the facility is public or private. Type of feedback, rank of feedback, and closure types are the demarcations assigned to the messages by the GRS system management team of MIS at DGHS. The type of feedback includes- complaints (e.g., long wait times, lack of doctors, inadequate facilities, etc.), compliments (e.g., satisfactory consultation services, good ambulance services, etc.), suggestions (e.g., recommendations to improve appointment scheduling systems, to improve storage facilities, to improve cleanliness of the hospital, etc.), and others (irrelevant messages). Feedback types are again subdivided based on the contents. After receiving each message, an assigned person in MIS confirms the nature of complaints by contacting the respective personnel in the complainant's facility. When complaints are found to be untrue, respective messages are assigned the term 'False' and stored. Although the subcategories can be reached in the graphical analysis part of the website, a detailed listing of these is unavailable from the "Messages" page of the website. The ranks are classified as major, moderate, and minor. These ranks are assigned by an expert health officer in MIS, based on a set of predetermined categorization guidelines and the officer's judgement on the imminence of the complaints and suggestions, and the level of satisfaction determined from the language of the compliments [10]. These rankings are frequently cross-validated through communication with respective authorities. The types of feedback that led to a form of "closure" includes resolved, forwarded, pending, overdue, and closed. The feedback is considered "resolved" when the matter in question is investigated and comes to a resolution. If the message requires forwarding to other departments of the health ministry for resolution, it is tagged as "Forwarded". Feedback is considered "pending" until an assigned person checks and takes the necessary action. If an MIS officer fails to take necessary steps against a message in time, it is considered "overdue". The feedbacks are marked "closed" when they cannot be verified or the steps of investigation cannot be followed [10].

### Data analysis

We retrieved and curated the data from GRS platform in Microsoft Excel 365. The closure or response time, i.e., the duration between message reception of the message and closure, was calculated by subtracting the date and time of the reception of the message from the date and time of closure and was converted into hours. Normality of the data was checked using a histogram. As the response time was highly skewed, we normalized the variable by taking the natural log, essentially assuming a log-normal distribution. Bivariate analysis was carried out using the Chi-square test for categorical variables. The Kruskal-Wallis test was used to compare the closure time data across groups of more than two categories. Post-hoc analysis was carried out using the Rank Sum test. The Kaplan-Meier Failure curve was generated to compare different types of closure across types of feedback. The log-rank test was done to statistically test the differences. Choropleth maps for the distribution of different types of feedback across districts were created using ArcGIS 10. The population data for each district was retrieved from the preliminary reports of the latest (2022) census by the Bangladesh Bureau of Statistics [11] for normalization of messages per million of the population of a district. Pie charts showing different categories and sub-types of messages, which are not available from the online tabular list, were retrieved from the graphical analysis page of the GRS dashboard [12]. All other data analysis was carried out in Stata Version 17.

### Result

The majority of the feedback was complaints (60.33%), followed by compliments (22.03%), and suggestions (14.48%). Of all feedback 31.46% was classified as major, 50.81% were moderate and 17.73% was minor. The responses to the feedback in descending order were forwarded (67.02%), closed (30.12%), resolved (2.55%), overdue (0.25%) and pending (0.05%). The median response time was 3.48 hours (IQR: 1.26 – 13.00, see Table 1).

The north-western and south-western districts of Bangladesh reported a greater number of messages per million citizen. The highest rates of the complaints (94–135 messages per million population) came from Jaipur Hat, Narail and Bagerhut. While the compliment rates were higher from Jashore, Satkhira, Chuadanga, Kushtia and Khagrachari (40 – 70

PLOS Digital Health

**Table 1. Feedback and response characteristics in the Grievance Record System.**

| Attribute | n (%) |
|---|---|
| **Type of feedback** | |
| Complaints | 6,977 (60.33) |
| Compliments | 2,547 (22.03) |
| Suggestions | 1,674 (14.48) |
| Uncategorized | 366 (3.16) |
| **Rank of feedback** | |
| Major | 3,598 (31.46) |
| Moderate | 5,810 (50.81) |
| Minor | 2,027 (17.73) |
| **Status of response** | |
| Closed | 3,494 (30.12) |
| Forwarded | 7,774 (67.02) |
| Resolved | 296 (2.55) |
| Overdue | 29 (0.25) |
| Pending | 6 (0.05) |
| **Response time (hours)** | |
| Median (IQR[a]) | 3.48 (1.26 – 13.00) |
| Min – Max | 0 – 770.53 |

Data source: Grievance Redress System (https://app.dghs.gov.bd/complaintbox/); [a]Inter Quartile Range.

messages per million population). Messages carrying suggestions predominantly came from Noagaon and Panchagarh (27–64 messages per million population) (Fig 1).

In terms of division (Level-2 administrative unit in Bangladesh), Rajshahi had the highest proportion of complaints (20.89%) and suggestions (32.30%), while Khulna had the highest number of compliments (15.70%). Rajshahi was also the division from which the highest number of messages were sent (22.44%, see S1 Table).

The GRS team in the MIS division of DGHS resolved only 3.27% of the complaints and forwarded 87.63% of the complaints. Compliments were mostly closed (96.86%), and suggestions were either closed (11.89%) or forwarded (86.44%) to respective divisional offices. Minor feedback was mostly closed (40.65%), and major feedback was mostly forwarded (73.57%). For feedback comprising complaints, those ranked 'major' were less likely to be resolved compared to moderate and minor ones (p < 0.001). Also, minor complaints were more frequently closed than other types of complaints (Table 2).

The majority of the complaints were related to the health workforce (26.7%), health infrastructure (25.9%), and service utilization (21.4%). Suggestions and compliments were mainly related to the health workforce (38.4% and 37.9%, respectively). The absence of existing service providers (52.0%) was the main complaint regarding health workforce; regarding health infrastructure, it was a waste management problem (56.2%); in justice and fairness-related complaints it was existence of outsiders/agents (44.1%); absence of basic amenities (75.0%) in relation to service readiness; and unavailability of essential medicines and medicine supplies under service utilization related complaints (34.7%) (Fig A-F in S1 File).

The median response time to complaints (3.58 hours, IQR: 1.34 – 13.36) was significantly higher compared to compliments (3.23, 1.20 – 9.81) and lower compared to suggestions (3.76, 1.23 – 14.67) (p < 0.001). Feedback classified as major (3.53, 1.41 – 12.95) and moderate (3.77, 1.24 – 14.09) needed a significantly higher response time compared to minor feedback (2.92, 1.18 – 8.57). Resolution of feedback (4.23, 1.58 – 14.14) needed a significantly higher response time compared to closure (3.43, 1.23 – 12.18) and forwarding process (3.50, 1.28 – 13.38) (p = 0.025). The response time

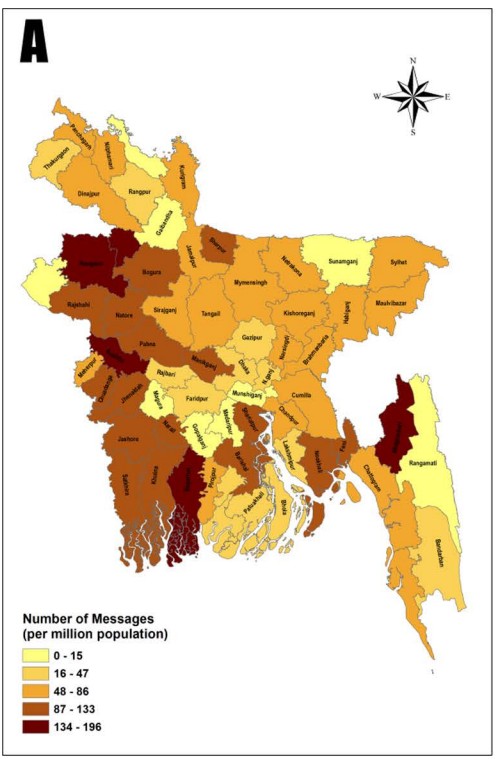
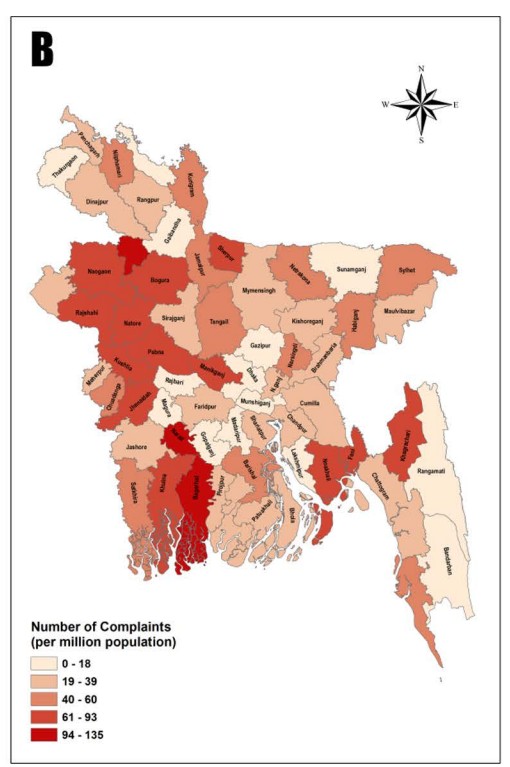
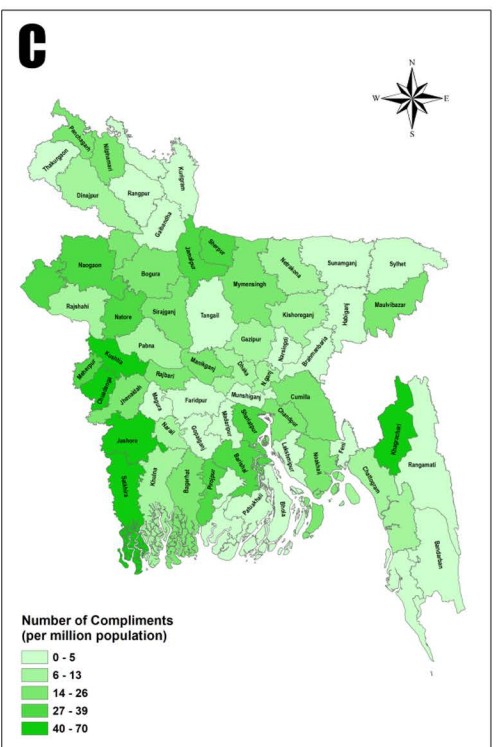
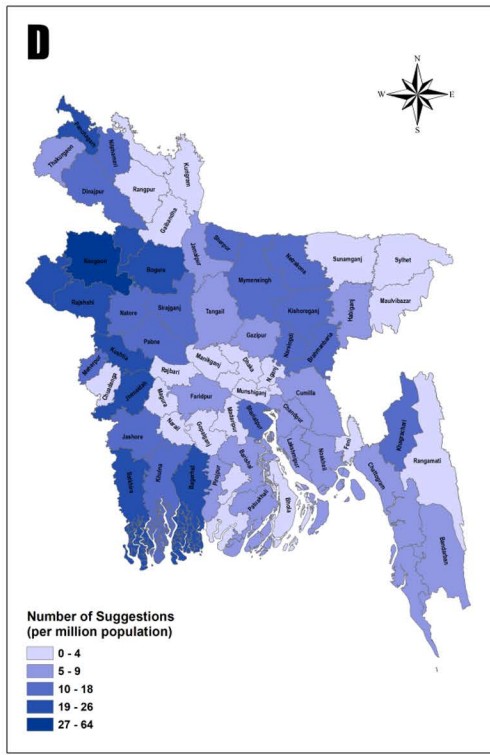

**Fig 1. District wise distribution of message (A), complaints (B), compliments (C) and suggestions (D).** [The figures were created using shape files freely available under Creative Commons Attribution 4.0 International License from https://data.humdata.org/dataset/cod-ab-bgd].

**Table 2. Relationship of feedback type and feedback rank with nature of response (status of response).**

| Variable | Nature of response | | | p-value |
|---|---|---|---|---|
| | Closed | Forwarded | Resolved | |
| **Feedback type** | | | | |
| Complaints | 635 (9.10) | 6,114 (87.63) | 228 (3.27) | <0.001 |
| Compliments | 2,467 (96.86) | 49 (1.92) | 31 (1.22) | |
| Suggestion | 199 (11.89) | 1,447 (86.44) | 28 (1.67) | |
| **Feedback rank** | | | | |
| Major | 872 (24.24) | 2,647 (73.57) | 79 (2.20) | <0.001 |
| Moderate | 1,758 (30.26) | 3,893 (67.01) | 159 (2.74) | |
| Minor | 824 (40.65) | 1,146 (56.54) | 57 (2.81) | |
| **Complaints rank** | | | | |
| Major | 192 (7.54) | 2,299 (90.30) | 55 (2.16) | <0.001 |
| Moderate | 272 (8.44) | 935 (87.66) | 42 (4.07) | |
| Minor | 160 (14.07) | 2,819 (82.23) | 131 (3.69) | |

Data within parenthesis represent row percentage; p-value determined by Chi-square test.

for minor complaints (3.91, 1.23 – 14.61) was significantly higher than for major complaints (3.55, 1.49 – 13.25) and moderate complaints (3.13, 1.34 – 8.71) (Table 3).

The Kaplan-Meier Failure estimates were calculated for time to resolution of different ranks of complaints (Fig 2). The estimates showed that the probability of resolution of minor and moderate complaints was significantly higher than major complaints (p<0.001 for both). The curve also shows that resolutions for all ranks of complaints tend to occur after more than two days (4 log normal hours = 54 hours).

**Table 3. Relationship of feedback type, feedback rank and status of response with response time.**

| Variable | Response time (hours) Median (IQR) | p-value |
|---|---|---|
| **Feedback type** | | |
| Complaints | 3.58 (1.34 – 13.36) | |
| Compliments | 3.23 (1.20 – 9.81)[a] | <0.001 |
| Suggestions | 3.76 (1.23 – 14.67)[b] | |
| **Feedback rank** | | |
| Major | 3.53 (1.41 – 12.95) | |
| Moderate | 3.77 (1.24 – 14.09) | <0.001 |
| Minor | 2.92 (1.18 – 8.57)[a,b] | |
| **Response status (overall)** | | |
| Closed | 3.43 (1.23 – 12.18) | |
| Forwarded | 3.50 (1.28 – 13.38) | 0.025 |
| Resolved | 4.23 (1.58 – 14.14)[a,b] | |
| **Complaint rank** | | |
| Major | 3.55 (1.49 – 13.25) | |
| Moderate | 3.13 (1.34 – 8.71) | 0.001 |
| Minor | 3.91 (1.23 – 14.61)[a,b] | |

P-value determined by Kruskal Wallis test; Post hoc analysis by Man-Whitney U test; Significant at p<0.05 level in relation to [a]first category and [b]second category.

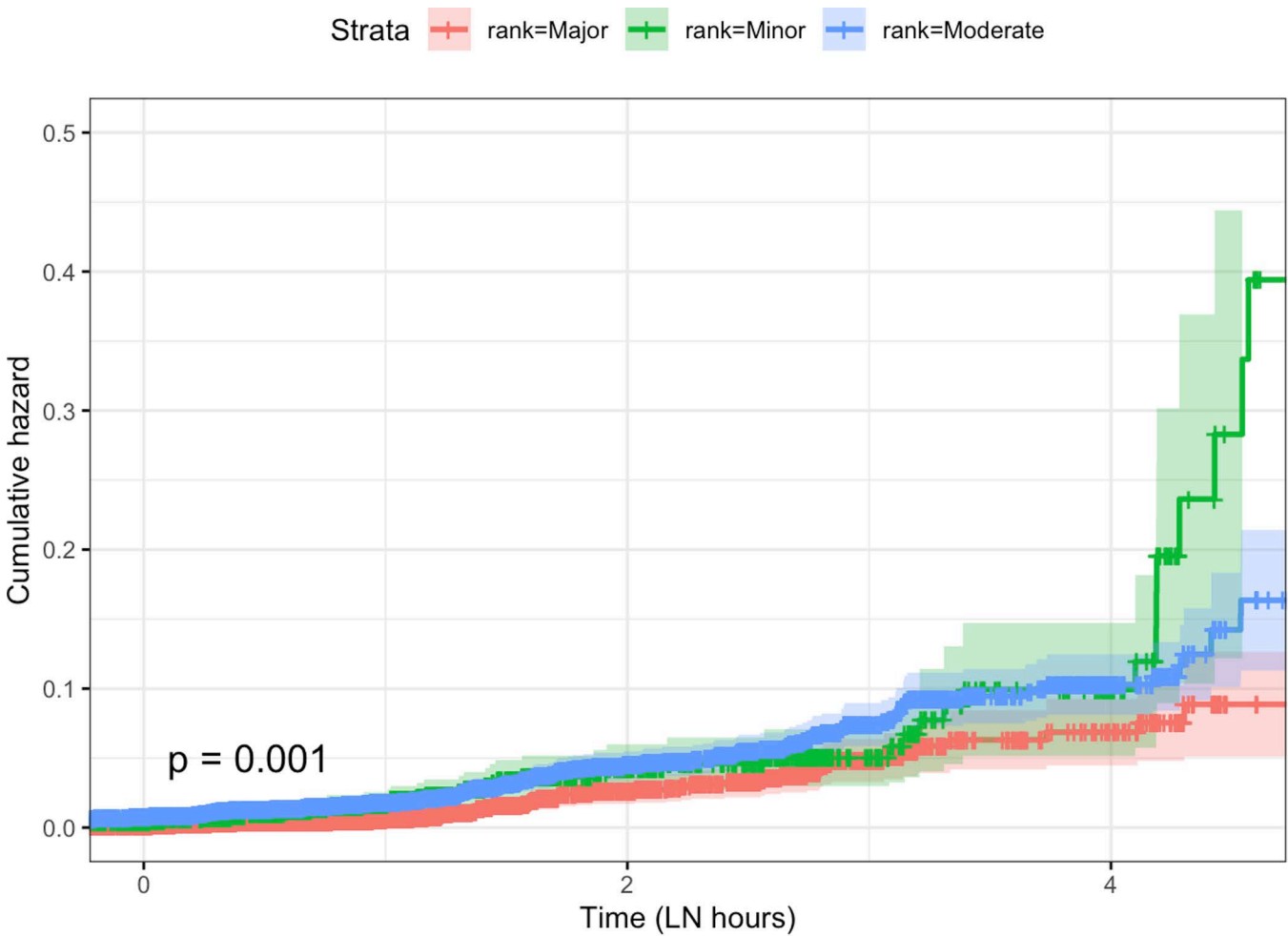

**Fig 2. Kaplan Meier Failure Curve showing the probability of resolving ranks of complaints (B) over time [Log-rank test with pairwise comparisons showed that the probability of resolving minor complaints was significantly higher than major complaints (p = 0.004), and the probability of resolving major complaints than moderate ones was significantly lower (p = 0.019).** There were no differences between moderate and minor resolution (p = 0.245)].

## Discussion

The GRS developed by MOHFW, Bangladesh, gives us a unique opportunity to assess the frequency, nature, and type of opinion of community people about the health care delivery system at the root level. The nature of problems faced at the service delivery points of the health system can be explored through public feedback. It also provides a platform to evaluate the responsiveness of health authorities to this feedback in the form of complaints and suggestions.

### Main findings of this study

Our study revealed that over a period of seven months, the GRS online portal received more than ten thousand messages. Of which, nearly two-thirds were complaints. The majority of complaints were related to the health workforce, health infrastructure, and service utilization. The absence of existing service providers, waste management problems, and the absence of basic amenities were the most prevalent complaints received. All of these complaints mirror the health

system constraints revealed by many previous studies [3,4,13–17]. The availability of a competent and adequate health workforce is a critical challenge faced by the health system of Bangladesh [3,15,16]. Ensuring a continuous supply of efficient health workers and their timely presence in the offices in a background of widespread corruption (10) had been a major challenge for many years [15]. Although the Government of Bangladesh (GoB) has a well-formulated health work-force strategy to tackle these issues [15], an appreciation of the issues arising at service delivery points through exploration, recognition, and resolution is required to ensure the presence of health workers in the facilities. For instance, one study revealed that doctors' attendance could be improved by various incentives, including functional securities, higher payments, a supportive environment, timely training, and promotion decisions [18]. However, disciplinary actions for absence are disliked by and considered a negative incentive by the doctors.

The absence of basic facilities, including essential medicines, diagnostic facilities, and logistic supply, has been a constraint in organizing non-communicable disease service delivery in primary health care [16]. Failure to handle the COVID-19 crisis at the initial stages was an eye-opener with respect to the persistent weakness and inadequacy of facilities for health delivery in developing countries like Bangladesh [13]. The feedback of the GRS system reveals the continuation of this problem in Bangladesh's health sector.

Improper management of medical waste, as reflected in the public opinions recorded by GRS, is another crucial problem in Bangladesh. A huge amount of waste is generated each day by hospitals. However, these are poorly managed. Poor medical management activities in district-level health facilities are evident from a recent study by Sujon and colleagues [19]. Waste is not segregated at the source or after collection. Improper waste collection, handling, storage, transportation, and disposal practices lead to various health hazards and environmental risks [20,21]. Although waste management strategies are outlined in the environmental assessment and action plan of the Health, Population, and Nutrition Sector Development Program (HPNSDP) [22] of Bangladesh, proper implementation of those strategies would require strict regulation, supervision, monitoring, and evaluation.

Our study reveals that there is a geographic variation in complaints, both in terms of volumes and rates. Healthcare seekers vary in their use of the GRS system in different districts and divisions. Therefore, prioritization and selective allocation of resources would help in the resolution and improvement of deficiencies in health care delivery.

Our analysis also revealed that the median response time for each type of message was reasonable, and complaints were given priority during resolution. However, the overall percentage of complaints resolved by the MIS, MOHFW seems very low. One reason for the low resolution is that the mechanism of retrieval of the final state of the resolution when a complaint is forwarded to other ministries is still not in place. Most of the 'major' complaints are forwarded to other departments of the health ministry or other ministries. But given the definitions of resolution [10], it is difficult to know from the GRS system how many of the forwarded complaints were resolved. Another reason for such a low resolution could be systemic bottlenecks, like the volume of feedback reaching the point of capacity saturation. However, as the GRS system is still being developed, the MIS has planned to increase the proportion of complaints being resolved by the department itself incrementally over the years. Nevertheless, our analysis revealed that a portion of all ranks of complaints are resolved. We found that minor and moderate complaints were more likely to be resolved than major complaints, resolutions tend to be slow, with the probability of resolution increasing after 54 hours had passed. This indicates a relatively sluggish response from the system, which should be expedited through necessary steps like training and posting of dedicated human resources for the system. The types of resolution are not specified in the GRS system. Hence, how the resolution occurs is not clear from the dashboard. As the system remains open to the public and may work as an important medium for improvement in public relations, the system should have this information presented in its graphical dashboard to increase transparency and demonstrate accountability.

## Assessment of the GRS system structure and recommendations

The GRS system structure stores and categorizes messages after their individual assessment by the assigned MIS officers. Currently, the process from complaint or compliment submission to resolution is not yet fully digitized. While individual

complaints are digitally tracked and subsequently revalidated by the Central Management Information System (MIS) team to ensure proper verification, the system primarily authenticates the complainant and assesses the source of the grievance. Internal documentation is used to categorize complaints by severity, which are then forwarded to the designated focal point at the respective facility. However, the subsequent action taken in response to the complaint—along with documented feedback from the focal person, typically the head of the institution—is not yet integrated into the same digital workflow. Efforts are currently underway to incorporate this final stage, which will complete the feedback loop of the digital GRS. Nevertheless, this is a tedious process where continuous training of the designated officers would be required, and an adequate supply of trained manpower needs to be ensured. Moreover, the probability of misclassification and ranking of messages due to subjective bias cannot be ignored. The system would be more informative when the closure types are further stratified based on follow-up information from other departments. However, this would require a cross-linkage with respective departments to ensure automated encoding of closure status and types. We recommend automation in the process of message classification, notification to specific departments, and closure status, thereby increasing the efficiency of the feedback system. This is possible through tokenization, classification, and sentiment analysis of message contents. However, this would require some preliminary studies to decipher the varieties of transcripts in which the messages are sent and find a way to make a uniform lexicon of words specific to public opinions regarding health services in the context of Bangladesh.

### Implications for practice

GRS is an innovative way to organize and manage a large amount of feedback coming from healthcare seekers as well as providers. This could be an excellent management solution to understand the needs, identify the gaps, and locate the shortcomings of the healthcare system of a country. It can reduce feedback management workload, inefficiency, and inco-ordination. However, enrichment of the system through the creation of a separate department with adequate manpower and resources would allow enhanced cooperation among ministries to provide comprehensive and sustainable management of the challenges faced by the healthcare system.

### Implications for research

The GRS system of the Directorate General of Health Services of Bangladesh creates an opportunity for various health systems and implementation research. Future research can explore many interesting areas, including whether GRS-based information can be used for differential allocation of manpower and healthcare resources, as a monitoring tool for quality healthcare, and as an evaluation metric for healthcare satisfaction at the user level.

### Implications for policy

The government of Bangladesh can take evidence-based strategies to enhance healthcare delivery using people's feedback from GRS. As the system records and categorizes complaints from health seekers with its a map of origin, it can prioritize and allocate resources to solve the problems of specific locations. This system can be adopted by countries around the world, particularly those with a large population and limited resources.

### Limitation and strengths

We had to limit our analysis of messages for the first eight months because of the unavailability of previous data on the freely accessible pages. An in-depth analysis of message contents was beyond the scope of this study. Nevertheless, this was one of the first studies using information from the GRS of the MOHFW of Bangladesh.

### Conclusion

The GRS is an important addition to evaluate health system structure and function through patient and provider feedback. However, its impact can only be fully appreciated through prompt, effective, and adequate responses to patient complaints

and suggestions. Compliments can be considered for providing incentives to healthcare workers working in the respective facilities. It can also be taken as a marker of improvement in healthcare delivery and infrastructure. As the nations pledged to work for universal health care, an effective grievance redress mechanism is essential to increase the quality of care. This study's findings could be considered by the health authorities to ameliorate the weaknesses of the existing health delivery system and strengthen and improve the online redress system for people's opinions regarding health services.

## Supporting information

**S1 File. Supplementary figures.**
(DOCX)

**S1 Table. Division wise distribution of feedback type.**
(DOCX)

## Acknowledgments

We are particularly indebted to MIS, DGHS for giving us permission to use GRS data and for the explanation regarding GRS maintenance process and for sharing the publicly available reports of GRS. We also cordially thank all the personnel involved in MIS for their relentless effort to ensure that GRS is running properly. Finally, we are grateful to the BRAC James P Grant School of Public Health and the CUNY Graduate School of Public Health & Health Policy for jointly organizing a training on population health informatics, as a part of which this research was conducted.

## Author contributions

**Conceptualization:** Md. Abdullah Saeed Khan, Atonu Rabbani.

**Data curation:** Md. Abdullah Saeed Khan, Md Toufiq Hassan Shawon.

**Formal analysis:** Md. Abdullah Saeed Khan, Atonu Rabbani.

**Methodology:** Md. Abdullah Saeed Khan, Atonu Rabbani.

**Project administration:** Md. Abdullah Saeed Khan, Md Toufiq Hassan Shawon.

**Resources:** Md. Abdullah Saeed Khan.

**Software:** Md. Abdullah Saeed Khan.

**Supervision:** Atonu Rabbani.

**Writing – original draft:** Md. Abdullah Saeed Khan, Atonu Rabbani.

**Writing – review & editing:** Md. Abdullah Saeed Khan, Md Toufiq Hassan Shawon, Atonu Rabbani.

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
