## [Decision Letter · Decision Letter 0]

Response to Reviewers
Revised Manuscript with Track Changes
Manuscript
**Journal Requirements:**

1. Please provide an Author Summary. This should appear in your manuscript between the Abstract (if applicable) and the Introduction, and should be 150–200 words long. The aim should be to make your findings accessible to a wide audience that includes both scientists and non-scientists. Sample summaries can be found on our website under Submission Guidelines

https://journals.plos.org/digitalhealth/s/submission-guidelines#loc-parts-of-a-submission

2. Figure 1: please (a) provide a direct link to the base layer of the map (i.e., the country or region border shape) and ensure this is also included in the figure legend; and (b) provide a link to the terms of use / license information for the base layer image or shapefile. We cannot publish proprietary or copyrighted maps (e.g. Google Maps, Mapquest) and the terms of use for your map base layer must be compatible with our CC-BY 4.0 license.

**Additional Editor Comments (if provided):**

1. The introduction extensively describes the context of globalization but does not adequately address the challenges facing Bangladesh’s health system. Although it concludes with 'we examined..., we explored...,' it fails to clearly articulate the research question, purpose, and significance of the study. It is recommended that the author include a clear statement explaining the study’s purpose and potential impact, such as its implications for public health systems, digital health platforms, or policymakers.

2. Page 5, lines 105–107: “Ranks include – major, moderate, and minor. These ranks are assigned by an expert of MIS based on the imminence of the complaints and suggestions, and the level of satisfaction determined from the language of the compliments.” The description of the classification criteria lacks clarity, potentially introducing a high degree of subjective bias. The author shall clarify whether the evaluation process incorporated mechanisms, such as a standardized operational manual, a dictionary, or a double-review process, to ensure consistency.

3. Page 5, lines 109–114: The manuscript does not adequately define the closure type. What is the difference between 'Resolved' and 'Closed'? Is 'Forwarded' equivalent to 'Unprocessed'? Additionally, is there any further follow-up? The author should provide a more precise definition and explanation.

4. The study used Kaplan-Meier survival curves and log-rank tests, using 'resolution' as the event of interest. However, the resolution rate was low (2.55%). Please reassess or justify the appropriateness of survival analysis in this context, or consider adopting an alternative approach.

5. Please expand the Discussion section (it is currently not there). The authors should consider including the following sections: implications for practice, implications for research, and implications for policy.

**Reviewers' Comments:**

**Comments to the Author**

1. Does this manuscript meet PLOS Digital Health’s publication criteria?

Reviewer #1: Yes

Reviewer #2: Yes

2. Has the statistical analysis been performed appropriately and rigorously?

Reviewer #1: Yes

Reviewer #2: Yes

3. Have the authors made all data underlying the findings in their manuscript fully available (please refer to the Data Availability Statement at the start of the manuscript PDF file)?

Reviewer #1: Yes

Reviewer #2: Yes

4. Is the manuscript presented in an intelligible fashion and written in standard English?

Reviewer #1: Yes

Reviewer #2: Yes

Reviewer #1: 1. Abbreviations must use at first place, multiple place this flaws found.

2. Provide examples of complaints, suggestions, and compliments to help contextualize the categories. For example:

Complaints: "Long wait times, lack of doctors, or inadequate facilities."

Suggestions: "Recommendations to improve appointment scheduling systems."

3. The percentages for response types (e.g., forwarding, closure, resolution) are useful, but a short explanation about why "resolution" is low (2.55%) could add value. For instance, are there systemic bottlenecks preventing resolution?

Reviewer #2: This is an important report to help keep the quality of services front of mind in Bangladesh. One question I had that might be included, does the geographic distribution of complaints relate in anyway to low resourced vs. high resourced areas?

**Do you want your identity to be public for this peer review?** For information about this choice, including consent withdrawal, please see our Privacy Policy

Reviewer #1: No

Reviewer #2: No

**Figure resubmission:****Reproducibility:** To enhance the reproducibility of your results, we recommend that authors of applicable studies deposit laboratory protocols in protocols.io, where a protocol can be assigned its own identifier (DOI) such that it can be cited independently in the future. Additionally, PLOS ONE offers an option to publish peer-reviewed clinical study protocols. Read more information on sharing protocols at https://plos.org/protocols?utm_medium=editorial-email&utm_source=authorletters&utm_campaign=protocols

---

## [Editor Report · Decision Letter 1]

Citizens’ Feedback on Health Service and the Responses of Health Authorities of Bangladesh: An Analysis of the Grievance Redress System

PDIG-D-24-00529R1

Dear Dr. Khan,

We're pleased to inform you that your manuscript has been judged scientifically suitable for publication and will be formally accepted for publication once it meets all outstanding technical requirements.

Within one week, you'll receive an e-mail detailing the required amendments. When these have been addressed, you'll receive a formal acceptance letter and your manuscript will be scheduled for publication.

An invoice for payment will follow shortly after the formal acceptance. To ensure an efficient process, please log into Editorial Manager at https://www.editorialmanager.com/pdig/ click the 'Update My Information' link at the top of the page, and double check that your user information is up-to-date. For billing related questions, please contact billing support at https://plos.my.site.com/s/.

Kind regards,

Calvin Or, PhD

Section Editor

PLOS Digital Health